# Digital Transformation of Health Professionals: Using the Context Optimisation Model for Person-Centred Analysis and Systematic Solutions (COMPASS) Implementation Model Use Case

**Carey Ann Mather** [1,*], **Joshua Fraser Bailey** [2] **and Helen Mary Almond** [3]

1 School of Nursing, College of Health and Medicine, University of Tasmania, Newnham 7248, Australia
2 School of Health Sciences, University of Tasmania, Newnham 7248, Australia
3 Australian Institute of Health Service Management, University of Tasmania, Hobart 7005, Australia
* Correspondence: carey.mather@utas.edu.au; Tel.: +61-3-6324-3149

**Abstract:** In today's demanding healthcare landscape, the use of theoretical frameworks is paramount for navigating the complexities of digital health challenges. The Context Optimisation Model for Person-centred Analysis and Systematic Solutions (COMPASS) theoretical framework and implementation model serves as an invaluable direction tool in planning, implementing, and evaluating digital healthcare initiatives. This paper showcases the tangible value of the COMPASS implementation model through a use case scenario involving an accredited exercise physiologist and a healthcare user with Type 2 Diabetes Mellitus who seeks credible information via a mobile digital device. Within this example, the COMPASS model demonstrates the ability to enhance systematic processes, streamline the workflow of health professionals and develop their capabilities to actively contribute to the transformative realm of digital health. Through exploration of the use case and the significance of the systematic processes as a research direction, the empowerment of health professionals to play pivotal roles in ongoing digital health transformation is emphasised. The COMPASS model emerges as a powerful tool, guiding health professionals and organisations towards innovative and sustainable solutions in the dynamic landscape of digital healthcare.

**Keywords:** framework; implementation model; digital; health; literacy; transformation; exercise physiologist; point of care; use case; healthcare user

## 1. Introduction

With health professionals reaching retirement age or choosing to leave healthcare delivery services, coupled with a rapidly expanding demand for healthcare services, healthcare users are experiencing the effects of the scarcity of qualified and experienced health professionals [1,2]. This exodus has increased the pressure on healthcare organizations to provide safe, quality care with fewer resources. Digital health transformation offers the ability to address some of these challenges by increasing efficiency and effectiveness, decreasing errors or near-misses, minimising waste, and enabling healthcare users to receive appropriate care. Automating repetitive tasks and streamlining workflows gives healthcare providers more time to promote digital literacy, person-centred care, and lessens administrative work [3].

Meeting the increased demand for safe, quality healthcare services presents significant challenges for the workforce as focus on human factors, rather than technology, is required [1]. Digital transformation can assist healthcare organizations recruit and retain a new generation of health professionals [3]. Embracing digital technology, such as mobile health (mHealth), electronic health records, and artificial intelligence, can provide career opportunities for increasing digital literacy and promote lifelong learning [4,5]. Digital transformation can also enhance the overall quality of care by enabling processes and

workflows where healthcare users receive best practice care, regardless of their condition, geographic location, or healthcare environment [6–9]. Globally, the COVID-19 pandemic exposed weaknesses in the healthcare workforce service delivery [7]. There is increased demand for digital health and technology educational preparation and the training of health professionals on how to manage evidence-based digital health transformation [10].

Frameworks can empower digitally capable health professionals to analyse healthcare events through evidence-based strategies and methods, enabling them to devise solutions for both routine and complex problems [11]. However, according to previous studies on digital capability frameworks [12,13], while they are useful in providing guidance for governments, organisations, and individuals in understanding the impacts of digital advancements, none of the reviewed frameworks included a specific domain for how to research and implement digital solutions in healthcare [12].

As part of the continuous improvement process, this article presents a use case applying an innovative theoretical framework and model developed to assist health professionals to approach digital transformation in a systematic way. Firstly, the implementation model is described. Next, the purpose of employment of the implementation model by health professionals and healthcare users at the point of care is outlined. Additionally, the context of the use case components is provided enabling an understanding of how the implementation model can be used within the context of a chronic disease management scenario. Section 2 provides the framework and boundary objects for the use case scenario. Section 3 demonstrates how the model can be implemented into health professional consultations to contribute to digital transformation. The implications of using the COMPASS implementation model at the point of care are then discussed. Lastly, strengths, limitations and future directions are indicated prior to the conclusion.

The Context Optimisation Model for Person-centred Analysis and Systematic Solutions (COMPASS) offers a theoretical framework and implementation model for increasing adoption of digital health solutions and capabilities in healthcare environments [12]. As a visual and narrative framework, COMPASS is a communication guide and mnemonic for a transformative digital health mindset that can assist healthcare workforces review any digital health issue in a systematic, person-centred manner. COMPASS includes characteristics that digitally capable health professionals need to consider before engaging with any individual or research actions outlined in Table 1 [12].

**Table 1.** COMPASS implementation model statements (modified from [12]).

| COMPASS Mnemonic Statements | |
|---|---|
| Context | Circumstance, situation, place, or time where digital transformation can influence healthcare safety and quality. |
| Optimisation | Methodologies adopted to ensure evidence-based therapeutic interventions that influence effective use of digital technologies to prevent, manage or treat health disorders or diseases. |
| Model | Plans, designs, implementations, and evaluations which best provide representations of digital technology within healthcare, as well as learning approaches that contribute to safety and quality. |
| Person-centred | Values, holistic lens, and inclusive goals facilitated by digital technology-enabled care, giving individuals authority to better engage with and control their health. |
| Analysis | Rigour, reliability, and validity ensuring detailed examination of digital transformations in healthcare to understand the nature or determine essential features. |
| Systematic | Methods required to develop and deliver a transformative approach in digital healthcare that is logical, repeatable, and able to be learned as an organised approach. |
| Solutions | Research impact, transformation, and future directions. Solutions resolve concerns using a transformative approach in digital healthcare. |

Planning, developing, implementing, analysing, and evaluating any digital health activity, quality assurance or research activity necessitates application of a systematic approach to guide the process. The COMPASS theoretical framework and implementation model was developed in stages and refined over 12 months. Firstly, a review of the literature was undertaken, and secondly expert review through a community of practices of digital health experts was sought. The theoretical framework and implementation model mnemonic was iteratively revised and finally validated by a community of practice participants [12]. COMPASS provides a theoretical framework and method for implementation to help individuals, groups, and organisations integrate digital health and contribute to workflows at all levels of healthcare [11].

For digital transformation to take place, health professionals need to make a concerted effort to support and enhance the digital and health literacy of healthcare users, enabling person-centred care [14,15]. Implementation of strategies to assist healthcare users to become competent in using digital health tools, such as mHealth, to maintain and promote health outcomes is necessary to contribute to effective and efficient healthcare service delivery [16]. Digital transformation needs all stakeholders to be digitally enabled [17,18]. Opportunistic educational opportunities for health professionals and healthcare users arise at the point of care, and therefore strengthening health and digital literacy needs to be harnessed during these consultations [16]. COMPASS offers evidence-based direction for health professionals to provide health education and promotion to healthcare users during consultations potentiating person-centred care into workflows in a systematic way [12]. Over time, at an individual level, this process will assist in the digital transformation of healthcare within healthcare environments and organisations.

Type 2 Diabetes Mellitus (T2DM) is a complex chronic condition that has a significant impact on the lives of those diagnosed with it [19]. This disease can be influenced by genetic and environmental factors and necessitates long-term management and monitoring. To mitigate the progression of T2DM, individuals need to adopt and maintain healthy lifestyle behaviours [20,21]. Such behaviours include consuming a balanced diet rich in vegetables, whole grains, and lean proteins, as well as engaging in regular physical activity and exercise [19].

In the jurisdictions of healthcare and complex chronic disease management, the concept of digital health is now prominent [1,12]. Digital health involves the application of digital transformation strategies, encompassing software and hardware solutions, to address various health and social care needs [12]. For health professionals, it is vital to recognise the lifestyle impacts of T2DM. This understanding forms the foundation for effective care and support. In today's healthcare environment, the value of digital transformation strategies cannot be overstated. Integrating digital health solutions allows health professionals to efficiently monitor and manage T2DM, enhancing the quality of care provided. In addition to addressing signs and symptoms, health professionals can use mHealth to encourage and track appropriate lifestyle behaviours, such as diet and exercise [9]. For healthcare users, these technologies equip individuals with the resources they need to achieve their long-term wellness goals. The synergy of traditional healthcare practices with digital transformation strategies can significantly improve overall care and outcomes for individuals with T2DM [12,22].

Accredited exercise physiologists (AEPs) are educationally prepared to assess health, undertake exercise assessment, design specialized exercise interventions, and guide physical activity and behaviour change [23,24]. They focus on developing tailored exercise plans to optimize the health of those with complex chronic diseases such as T2DM [25], enhancing functional recovery, promoting health and well-being and facilitating the independence of healthcare users [23]. AEPs are accredited to design and evaluate active therapies, and their scope of practice includes therapeutic interventions to monitor and treat healthcare users. Within an integrated healthcare team, AEPs are well placed to adopt and engage with mHealth to implement person-centred care through strengthening the digital literacy of healthcare users during consultations at the point of care [2,14,24]. Additionally, rap-

port building between health professionals and healthcare users can be developed during interactions to promote trust and continuity of care [14,24].

Management of T2DM in conjunction with AEP consultations has demonstrated positive benefits for the incorporation of exercise as part of T2DM management [9,19]. Interaction between a T2DM healthcare user and an AEP has been selected to present a person-centred use case applying the COMPASS theoretical framework and implementation model. This use case illustrates the application of a systematic process when integrating digital technology into healthcare workflows. Sustained emphasis on digital health transformation provides the opportunity for those diagnosed with T2DM to enhance self-efficacy in both digital and health literacy, promoting adoption of long-term positive well-being behaviours [26]. Components of this use case can be implemented and assessed for usability across a spectrum of health professional roles. Person-centred consultations incorporating education about digitally accessible credible information has the potential to enhance workflows and foster positive impacts on health, well-being, and digital health.

## 2. Materials and Methods

The COMPASS theoretical framework and implementation model, as described by Mather and Almond [12], was applied to a use case example illustrating the planning, development, analysis, and implementation of digital health interventions. T2DM was chosen for this use case due to its complex nature as a chronic condition, necessitating a person-centred approach involving a range of health professionals. Exercise interventions are widely recognised for their role in maintaining and improving health and overall well-being [19]. AEPs are trained to assess health, perform exercise assessments, create customized exercise programs, and guide changes in physical activity and behaviour [22,23].

The presented use case demonstrates the utility of the COMPASS implementation model and aims to showcase how it can be applied to navigate complex digital and healthcare scenarios. In this use case, clear and shared definitions are vital. In healthcare, technology, social aspects, and physical objects have served as tools that enable multiple stakeholders to interpret interactions in their own way when collaborating on new ideas and improvements [27]. Table 2 represents the boundary objects that facilitate a shared understanding among health professionals of the context and conditions in the use case scenario, enabling interpretations from various perspectives [28,29]. Individual components of the use case and how the COMPASS implementation model can be applied are shown in Table 3, providing information about the conditions required for successful delivery of the use case scenario. Table 1 provides the COMPASS implementation model mnemonic statements [12] and is applied to the T2DM use case scenario demonstrated in Figure 1. Figure 1 shows how COMPASS can be applied to the individual workflow of an AEP and healthcare user with T2DM [25].

**Table 2.** Boundary objects for COMPASS use case scenario.

| Scope | Duration | Participants | Props | Trigger |
|---|---|---|---|---|
| **Start:** Accredited exercise physiologist (AEP) engages healthcare user **End:** The AEP develops a rapport, gains trust, and demonstrates capability in supporting the healthcare user in finding credible information using a mobile tablet device | Approximately 30 min | Healthcare user AEP | Mobile tablet device internet/access to web-based resources | Healthcare user has to express interest in learning about T2DM, particularly with regard to exercise and the management of signs and symptoms |

**Table 3.** Pre- and post-conditions for the use case context.

| Conditions |
|---|
| **Pre-conditions** |
| Healthcare user is empowered to understand digital technology and has the capability and capacity to seek information. |
| The AEP has transitional or entrustable digital health capabilities to support the healthcare user's request. |
| The AEP and healthcare user have access to a mobile device and wireless internet at the point of care. |
| **Post-conditions** |
| The healthcare user will know how and where to access credible information on the internet and what they need to know about monitoring and undertaking exercise between AEP consultations. |
| The AEP will have responded to the digital health literacy development of the healthcare user by sharing how to discern credible information, showing the healthcare user how to browse for credible websites. |
| The AEP will have gained an understanding concerning the well-being of the healthcare user and their digital literacy. |
| The AEP and healthcare user will have a shared understanding about how they can both monitor T2DM, specifically undertaking exercise while managing well-being. |

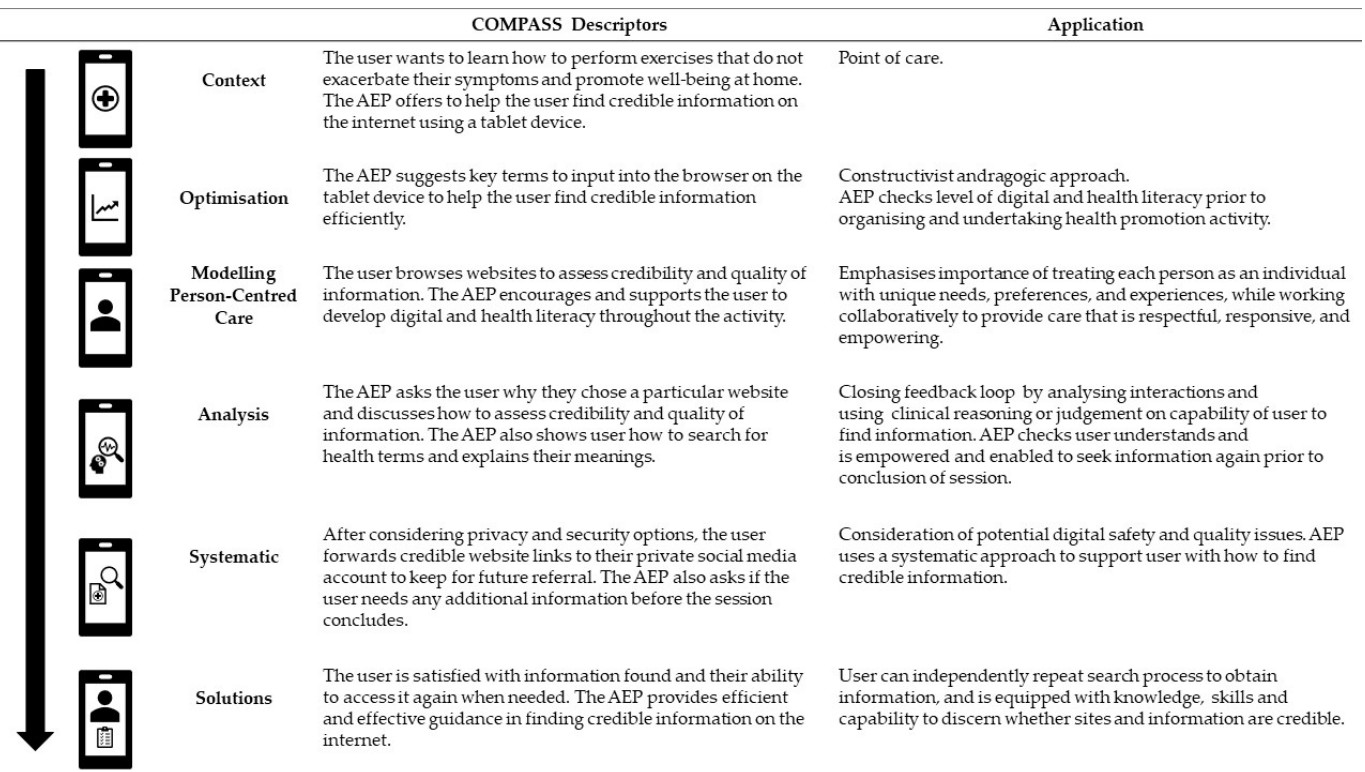

| | COMPASS | Descriptors | Application |
|---|---|---|---|
| | Context | The user wants to learn how to perform exercises that do not exacerbate their symptoms and promote well-being at home. The AEP offers to help the user find credible information on the internet using a tablet device. | Point of care. |
| | Optimisation | The AEP suggests key terms to input into the browser on the tablet device to help the user find credible information efficiently. | Constructivist andragogic approach. AEP checks level of digital and health literacy prior to organising and undertaking health promotion activity. |
| | Modelling Person-Centred Care | The user browses websites to assess credibility and quality of information. The AEP encourages and supports the user to develop digital and health literacy throughout the activity. | Emphasises importance of treating each person as an individual with unique needs, preferences, and experiences, while working collaboratively to provide care that is respectful, responsive, and empowering. |
| | Analysis | The AEP asks the user why they chose a particular website and discusses how to assess credibility and quality of information. The AEP also shows user how to search for health terms and explains their meanings. | Closing feedback loop by analysing interactions and using clinical reasoning or judgement on capability of user to find information. AEP checks user understands and is empowered and enabled to seek information again prior to conclusion of session. |
| | Systematic | After considering privacy and security options, the user forwards credible website links to their private social media account to keep for future referral. The AEP also asks if the user needs any additional information before the session concludes. | Consideration of potential digital safety and quality issues. AEP uses a systematic approach to support user with how to find credible information. |
| | Solutions | The user is satisfied with information found and their ability to access it again when needed. The AEP provides efficient and effective guidance in finding credible information on the internet. | User can independently repeat search process to obtain information, and is equipped with knowledge, skills and capability to discern whether sites and information are credible. |

**Figure 1.** The COMPASS implementation model use case scenario.

## 3. Results

Figure 1 provides descriptors for the mnemonic for understanding the COMPASS implementation model and its practical application within a complex use case. This use case illustrates a systematic process of integrating digital technology [2]. Implementation of efficient workflows yields a two-fold benefit: it allows health professionals to allocate more time to delivering person-centred care, and it supports ongoing development of a new generation of health professionals within healthcare organisations [3]. Figure 1 also demonstrates how the COMPASS theoretical framework can be employed to direct the workflow of this AEP model, enabling a person-centred approach to healthcare delivery. This systematic approach facilitates the creation of a comprehensive, relevant, and per-

sonalized care plan that addresses the complexity of the healthcare user's health status, knowledge gaps, and pertinent risk management information related to their T2DM.

In context, the AEP establishes a rapport, fosters trust, and promotes the healthcare user's health and digital literacy. This is achieved by empowering the healthcare user to access trustworthy information about T2DM through the use of mHealth to review their lifestyle objectives [2]. Furthermore, this approach allows the AEP to gauge the healthcare user's health and digital literacy knowledge and skills, enabling person-centred future follow-up and equitable access to contemporary and relevant healthcare information.

## 4. Discussion

This use case focuses on how to use the COMPASS implementation model to increase digital health adoption in healthcare [12] for individual activities. It also offers guidance for learning by healthcare organizations. Each contribution has the potential to foster and modify an individual digital health scenarios towards becoming normal behaviours [24,30]. Other components of the COMPASS theoretical framework and implementation model can be used to further incorporate processes that are described elsewhere [12]. Advantages of using the COMPASS implementation model include offering a structured approach to drive digital health transformation on individual, system, and ultimately organizational scales. Kennedy and colleagues [31] highlighted the benefits of delivering integrated services, which included AEPs working with people receiving complex treatments, demonstrating tangible advantages for healthcare users. Kennedy and colleagues [31] recommended implementation plans as essential for the sustainability of integrated care that included AEPs. The feedback received from the participants regarding the role of AEPs reported that supervised exercise tailored to them was beneficial to their treatment experience and well-being [8,31]. The participants also reported a sense of positivity and opportunities to build a rapport and relationships with other healthcare users. Kennedy and colleagues [8] indicated that the incorporation of theoretical frameworks and implementation models, such as COMPASS, into workflows can assist the promotion of support to healthcare users between AEP visits and empower healthcare users to build their own support networks. The COMPASS implementation model was designed to assist in addressing digital health challenges at organizational or system levels. It provides a systematic process to direct health professionals towards enhanced efficiency and effectiveness in delivering appropriate safe healthcare interactions [12]. Harrison and colleagues [11] suggested that such change management methodologies are useful in guiding improvements in complex healthcare contexts; however, they found a lack of application to support the implementation [11].

The growth of digital technologies to strengthen healthcare requires workforces to be health and digitally literate [13,32]. The COVID-19 pandemic accelerated the use of digital technologies within healthcare environments [33], creating opportunities for the development of the COMPASS theoretical framework and implementation model to support an unmet need in digital health [12,14,24]. To increase the demands of healthcare needs during the COVID-19 pandemic, health professionals were required to pivot, upskill, and deliver care in innovative ways, which required the use of digital health and embedded technologies. These skills have remained part of the 'new normal' as the healthcare pandemic moves towards epidemic management [32,34,35]. However, theoretical models and implementation frameworks supporting these requirements lag behind, further increasing the perceived workload for an already over-burdened workforce [1].

This use case demonstrates how point-of-care interactions can continuously support a positive healthcare user experience, as reported by Kennedy and colleagues [31]. The incorporation of digital technologies to support the monitoring and management of chronic diseases is escalating [7]. Ensuring a person-centred approach, where healthcare users are educationally prepared and capable of monitoring or managing their own care at home is empowering and economically resourceful [6,7]. Using the COMPASS theoretical framework and implementation model [12] as a person-centred model reinforces concepts using digital technology at the point of care by healthcare teams, empowering healthcare

users to be independent between healthcare consultations [6]. This use case demonstrates how a systematic approach to the implementation of digital technology into individual point-of-care interactions can strengthen and support safe quality healthcare.

## 5. Strengths and Limitations

COMPASS is a theoretical framework and implementation model that needs to be used systematically. The strengths of the COMPASS theoretical framework and implementation model were developed iteratively over a 12-month period with feedback from international digital health experts. As a new framework and implementation model, it needs to be rigorously tested and its usability reported in a range of contexts. A limitation of the implementation model is it assists with how to solve issues; it does not solve healthcare issues. The model relies on robust input from stakeholders. If erroneous judgments are included at any stage or sections of the process are overlooked, the implementation model described here may not achieve the desired solutions. Continuous feedback on its usability will be used to strengthen the COMPASS framework and implementation model to support future digital transformation within healthcare contexts.

## 6. Future Directions

This use case was presented through the lens of an AEP and healthcare user with T2DM. Further research engaging with multiple health professional roles delivering care to a diverse range of healthcare users in a variety of healthcare environments is required to explore the scope and value of COMPASS. Further usability testing of the components of the theoretical framework and implementation model are desirable to confirm it remains up to date within the rapidly evolving landscape of healthcare and digital technologies.

## 7. Conclusions

Models can be used to guide healthcare users on how to present or consider options. The COMPASS theoretical framework and implementation model provides a direction using a systematic process to assist in solving simple and complex healthcare questions at individual, organisation, or system levels. The COMPASS use case demonstrates how implementation models can be applied to workflows at the individual level, which contributes to digital transformation at all levels. It promotes the implementation and acceptance of digital technology by workforces engaged in developing and delivering healthcare.

**Author Contributions:** C.A.M., J.F.B. and H.M.A. contributed to the conceptualisation, methodology, analysis, and original draft preparation. All authors have read and agreed to the published version of the manuscript.

**Funding:** This research received no external funding.

**Institutional Review Board Statement:** Not applicable.

**Informed Consent Statement:** Not applicable.

**Data Availability Statement:** The COMPASS theoretical framework and implementation model is published and available for use. Please contact the corresponding author for further information.

**Conflicts of Interest:** The authors declare no conflict of interest.

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
