# Peer review of "Digital Transformation of Health Professionals: Using the Context Optimisation Model for Person-Centred Analysis and Systematic Solutions (COMPASS) Implementation Model Use Case"

_knowledge, doi:10.3390/knowledge3040042_

Round 1
Reviewer 1 Report
Comments and Suggestions for Authors
In this paper, the authors delve into the digital transformation of healthcare professionals through the use of the COMPASS implementation model as a case study. The COMPASS theoretical framework and implementation model offer a systematic approach to address healthcare questions, whether simple or complex, at individual, organizational, or system levels. However, there are several areas of concern in this paper:
1. The title inconsistently uses abbreviations and full names, and it contains some grammar mistakes that need correction.
2. The language quality should be refined, as most journals adhere to American-style English. Attention should be paid to the spelling of "organization" to align with this style.
3. Both the abstract and Section 1 lack a strong motivation for presenting COMPASS, and the logic of the literature review is weak. Consider adding a separate section for related works and summarizing the main contributions.
4. The quality of Figure 1 is subpar, and more illustrations should be included to enhance clarity.
5. Sections 3-5 could potentially be consolidated, and the presentation of results and their implications is not sufficiently clear.
6. The reference list should be updated to maintain a consistent style and incorporate recent works.
7. There are numerous abbreviations in this version. It is essential to create a dedicated table to compile and clarify these abbreviations for better understanding.
Comments on the Quality of English LanguageModerate editing of English language required.
Reviewer 2 Report
Comments and Suggestions for Authors
This research topic is interesting and valuable. However, there are many places that need to be revised and improved.
(1) I believe the keywords can be reduced to at most five important words.
(2) The details of the analysis results should be complete. Please include more contents.
(3) The limitation and future directions can be combined.
(4) The conclusion section can be put before the limitation section and should include the policy suggestions.
Reviewer 3 Report
Comments and Suggestions for Authors
The paper entitled "Digital transformation of healthcare professionals: Using COMPASS (Context Optimisation Model for Person-Centred Analysis and Systematic Solutions) implementation model use case" focuses on the assessment of the practical application of COMPASS implementation model using a scenario of a patient with Type 2 Diabetes Mellitus seeking credible information via a mobile digital device. The paper is well-written and conveys a clear message. It employs non-trivial tools to produce relevant results and draw implications and conclusions. Overall, I enjoyed reading this paper and I have just a handful of comments and suggestions for the authors:
1. In the Introduction, the authors might want to discuss COMPASS in greater detail and add more relevant information and sources apart from just one paper by Mather and Almond (2022). What about other applications and usage?
2. The value-added of this paper needs to be highlighted as well as its limitations. In addition, it would be better to present the structure of the paper in the Introduction.
3. In its essence, COMPASS is about telemedicine, so more literature review on this field might be added.
4. Figure 1 contains very small print that is difficult to read. I would be better to re-do the figure in a form of a flow diagram or a dynamic chart.
5. The Conclusions should be written up in order to include relevant policy implications and a more thorough discussion on the use of models such as COMPASS in healthcare (I would recommend mentioning the telemedicine and commenting on its raise and popularity in the recent years). What about the role and the place of various smart watches and other wearable devices? This might be an interesting topic for discussion within the context of this paper.
Round 2
Reviewer 2 Report
Comments and Suggestions for Authors
Great, no more comments!